# Damage Characteristics and Mechanism of the 2017 Groundwater Inrush Accident That Occurred at Dongyu Coalmine in Taiyuan, Shanxi, China

**Bin Luo, Yajun Sun \*, Zhimin Xu, Ge Chen, Li Zhang, Weining Lu, Xianming Zhao and Huiqing Yuan**

School of Resources and Geosciences, China University of Mining and Technology, Xuzhou 221116, China; luobin@cumt.edu.cn (B.L.); xuzhimin@cumt.edu.cn (Z.X.); cg5@cumt.edu.cn (G.C.); zhangli2010cugb@163.com (L.Z.); Luweining1111@163.com (W.L.); tb19010023b0@cumt.edu.cn (X.Z.); ts18010098a31@cumt.edu.cn (H.Y.)
* Correspondence: syj@cumt.edu.cn; Tel.: +86-13912005180

**Abstract:** On 22 May 2017, a groundwater inrush accident occurred in the gob area of coal floor at Dongyu Coal Mine in Qingxu County, Shanxi Province, China. The water inrush accident caused great damage, among which six people died and the direct economic loss was about CNY 5.05 million. An elliptical permeable passage appeared at the floor of the water inrush point, and the lithology of the outburst is mainly fragmented sandy mudstone and siltstone of coal roof No.2 in the lower layer of coal seam No.3, which is currently being mined, with a peak inflow of 500 m³/h. The water inrush happened due to following reasons: There is an abandoned stagnant water-closed roadway in coal seam No.2, which is the lower mine group of coal seam No.3. The abandoned roadway of coal seam No.2 is an inclined roadway. The water level of the roadway far away from the accident point is higher than the floor elevation of coal seam No.3. Under the joint action of water pressure, mining disturbance, and weakening of goaf water immersion, the original equilibrium state was broken, resulting in the destruction of the only 7 m water-barrier rock pillar between coal seam No.3 and coal seam No.2. The water in the goaf led upward along the roof crack, gradually evolved from seepage to gushing water, and a large amount of goaf water poured into the roadway in the working face of the 03304 panel, finally leading to the occurrence of catastrophic water inrush. Technically, the miners did not implement the technical provisions of the coal mine water control regulations, leading to the accident. In addition, the failure to arrange evacuees to a safe location after apparent signs of water inrush also increased the catastrophic level of the accident.

**Keywords:** groundwater inrush; Dongyu coalmine; damage mechanism; goaf water accidents

## 1. Introduction

Underground tunnel excavation and coal mining processes have obvious influence on the nearby strata. At the same time, nearby hydrogeological conditions change accordingly. As a result, the phenomenon of groundwater flooding into mining areas occurs from time to time all over the world. Cases in Poland, Australia, Britain, Italy, China, India, and other places have been reported, and the reasons have been discussed [1–5]. Finally, in order to reduce their great harm, scholars from various countries put forward many effective methods [6–8].

In China, coal production has stabilized at a high level to meet the huge demand of the near-industrial economy. Water inrush sometimes occurs in coal mines under the background of high production capacity. Figure 1 details some properties of water inrush accidents from 2000 to 2020; there were a total of 779 water inrush accidents and 3831 deaths, among which 527 occurred in goaf water inrush accidents and 2936 people died. The number of water inrush accidents in goaf accounted for 67.7% of the total, and the death toll was as high as 76.6%. On the whole, the number of accidents and the number of

deaths have been significantly reduced, but the proportion of goaf water inrush accidents is always high and has brought great damage [9–11]. Shanxi is the province most prone to old goaf water inrushes, where 8780 coal mines have been abandoned in the last 20 years [12]. Therefore, there is still a long way to go in the study of goaf water inrush accidents.

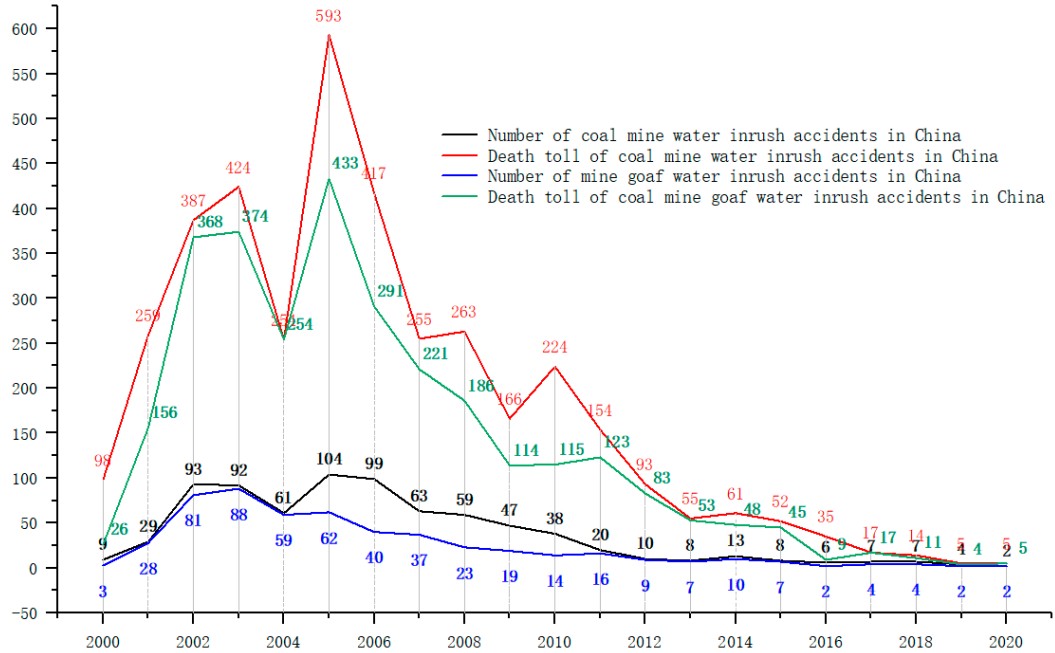

**Figure 1.** Details of some properties of water inrush accidents from 2000 to 2020 in China.

In fact, scholars and engineers have conducted a lot of related research in China. Early exploration, prediction, evaluation of geological conditions and types of water disasters, and methods of prevention and control are the focus of these studies [13–21]. All of these effectively reduce the occurrence of water inrush accidents in coal mines.

Unfortunately, at 23:38 hours on 22 May 2017, a water inrush accident occurred at Dongyu Coal Mine in Qingxu County, Shanxi Province. When the accident happened, 11 miners were trapped underground in the coal mine. Fortunately, five miners were rescued from the accident, but six miners drowned. Furthermore, the accident caused direct economic losses of over CNY 505 million.

## 2. Materials and Methods

The coal mine goaf is densely distributed in Shanxi, China, which causes many water inrush accidents. Figure 2 shows the distribution of typical water inrush accidents in the goaf of Shanxi Province, and Table 1 shows the specific names and accident characteristics.

Located in Taiyuan (Figure 2), Shanxi, construction of Dongyu Coal Mine began in 2011, and it was put into operation in 2015. Its annual design capacity was 1.5 million tons of raw coal. The coalfield is about 5.15 km long in the north–south direction and 6.25 km wide in the east–west direction. Its area is about 16.85 km². The coal reserves are estimated at about 0.14 billion tons, and the exploitable amount is about 0.27 billion tons.

The northeast part of the mine is adjacent to Niangou Coal Mine; the east part is adjacent to Lijialou Coal Mine; the southeast part is a goaf without mining rights; the southwest part is adjacent to Ruize Coal Mine; the west part is adjacent to Nanling Coal Mine; and the north part is adjacent to Zhaojiashan Coal Mine. In addition, there are three closed coal mines merged into Dongyu Coal Mine, as shown in Figure 2.

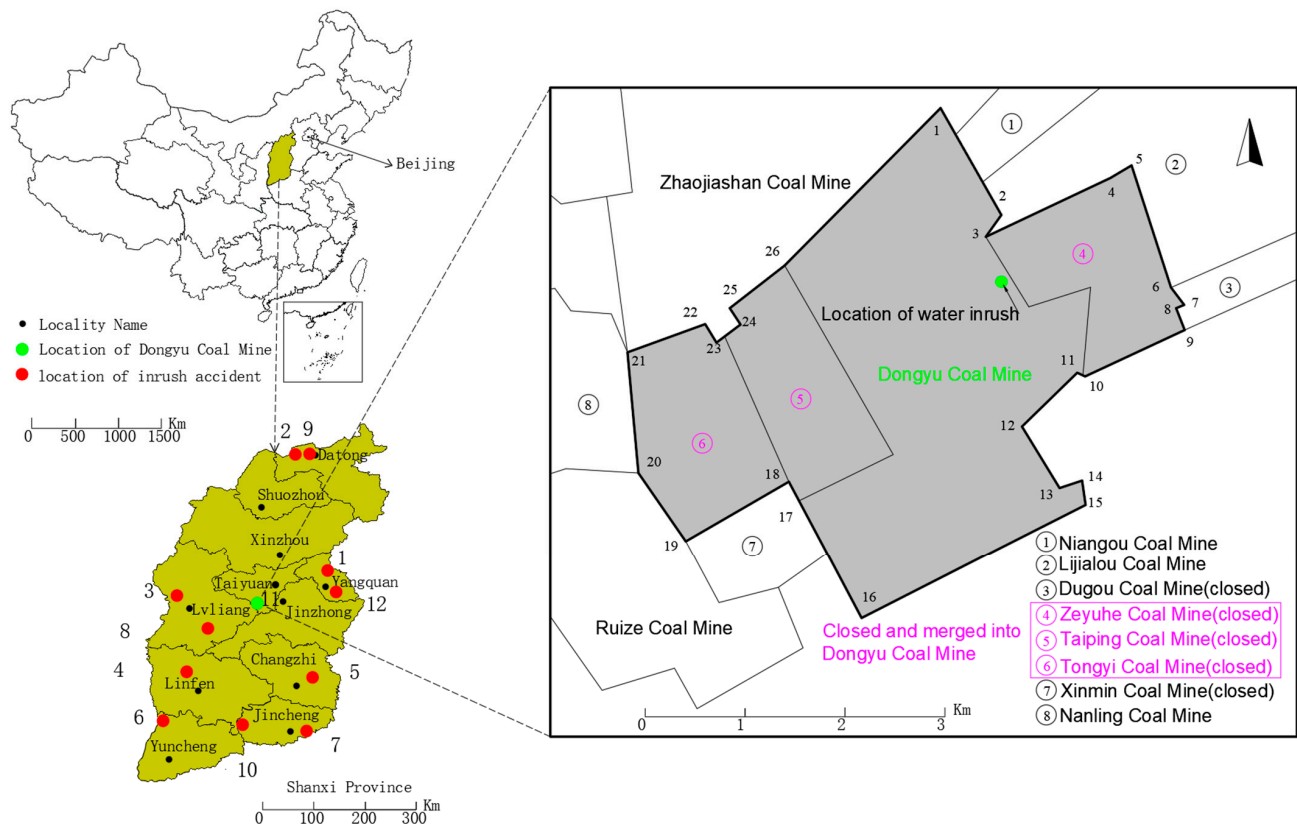

**Figure 2.** Distribution of 11 water inrush accidents in Shanxi and location relation diagram of adjacent mines of Dongyu coal mine.

**Table 1.** Statistics of 11 major air water inrush accidents in Shanxi Province.

| No. | Time | Location | Death Toll | Coal (Rock) pillar/m |
|---|---|---|---|---|
| 1 | 1992.11.05 | Shenyigou Coal Mine in Yangquan City | 51 | 1.5 |
| 2 | 2005.05.18 | Xinjing Coal Mine in Zuoyun County | 56 | 1 |
| 3 | 2006.03.18 | Fanjiashan Coal Mine in Linxian County | 28 | 2 |
| 4 | 2008.02.24 | Donghe Coal Mine in Linfen City | 7 | <6 |
| 5 | 2010.02.21 | Zhangcun Coal Mine in Changzhi City | 1 | 7 |
| 6 | 2010.03.28 | Wangjialing Coal Mine in Xiangning County [9] | 38 | 0.8 |
| 7 | 2010.06.03 | Jiaonan Coal Mine in Jincheng City | 5 | 2 |
| 8 | 2013.09.28 | Zhengsheng Coal Mine in Fenyang City | 10 | 1 |
| 9 | 2013.11.29 | Xingwang Coal Mine in Datong City | 4 | 1 |
| 10 | 2016.07.02 | Zhongcun Coal Mine in Jincheng City [22] | 4 | 1.8 |
| 11 | 2017.05.22 | Dongyu Coal Mine in Qingxu County | 5 | 7 |
| 12 | 2017.07.31 | Weifeng Coal Mine in Pingding County | 5 | 2 |

## 3. The Geological Setting

The Dongyu coalfield is located in the South East of Xishan coalfield, between the south side of the Niandi fault and the northwest side of the Qingjiao fault. The annual precipitation is 249.5–495.7 mm, and the annual evaporation is 1715.6–2047.6 mm. There are four gullies running from northwest to southeast, which are seasonal flood discharge gullies with less water at ordinary times.

The surface bedrock in the mine field is well exposed, and the Quaternary middle upper Pleistocene loess is scattered over the gully slope and ridge. As shown in Figure 3, the strata formed in the coalfield in chronological order include the following: (1) karst limestone formed in the Ordovician period; (2) coal seam, west grain sandstone, siltstone, sandstone mudstone, mudstone, and limestone formed in the Carboniferous period; (3)

coal seam, coarse-grained sandstone, medium grained sandstone, fine-grained sandstone, sandy mudstone, mudstone, and carbonaceous mudstone formed in the Permian period; (4) loess, pebbles, and gravels formed in the Quaternary period. The thickness of Quaternary Middle–Upper Pleistocene ($Q_{2+3}$) is 0–32.00 m, with an average of 6.00 m. It is in angular unconformity contact with the underlying strata. The thickness of Quaternary Holocene ($Q_4$) is 0–5.00 m, with an average of 2.00 m. It is mainly distributed in the southeast valley of the minefield and is alluvial formed by sand, pebbles, and rock blocks of various rock components.

| strata system | average thickness /m | depth /m | stratigraphic column | lithology | aquifers |
|---|---|---|---|---|---|
| strata system | 8.00 | 8.00 | | sand bed & gravel | porous phreatic aquifer |
| | 285.38 | 293.38 | | Interbedded medium-coarse sandstone sandy mudstone Interbedded medium fine sandstone | impermeable layer |
| | 14.62 | 308.00 | | coarse sandstone($K_6$) | fractured confined aquifer |
| | 113.97 | 421.97 | | sandy mudstone silty sandstone sandstone and mudstone interbed coarse sandstone mudstone | impermeable layer |
| Permian | 4.64 | 426.61 | | luotuobozi sandstone（K4） | fractured confined aquifer |
| | 14.55 | 441.16 | | No. 03 coal seam | location of the water inrush |
| | 13.70 | 454.86 | | No. 2 3 ∞ 4 coal seam | water inrush source |
| | 25.35 | 480.21 | | No. 5 coal seam moderate coarse sandstone | fractured confined aquifer |
| Carboniferous | 56.10 | 536.31 | | No. $6^{-1}$ 6 ∞ 7 coal seam siltstone $L_1$、$L_4$ & $L_5$ limestone carbonaceous mudstone No. $8^{-1}$ 8 ∞ 9 coal seam | fractured confined aquifer |
| | 33.64 | 569.95 | | sandy mudstone Jinci sandstone No. 10 ∞ 11 coal seam | impermeable layer |
| | 33.00 | 602.95 | | mudstone Sandy mudstone Aluminum mudstone | impermeable layer |
| Ordovician | 120.00 | 722.95 | | karst limestone | karst confined aquifer |

**Figure 3.** Strata and aquifer distribution at Dongyu coal mine.



Coal seams in the Dongyu coalfield are located mainly in the Permian and the Carboniferous strata. Coal seams No.3, No.2, No.4, No.5, No.8, and No.9 are the main minable ones. Their total thickness is about 16.21 m. Their properties are detailed as follows: the thickness of coal seam No.3 coal from 1.00 to 2.30 m, and the average is about 1.68 m. Its roof is composed of mudstone, sandy mudstone, siltstone, and medium-fine sandstone. Its floor is composed of mudstone, sandy mudstone, and medium-fine sandstone. The average thickness of coal seam No.2 is about 2.45 m. Its roof is composed of carbonaceous mudstone, mudstone, sandy mudstone, and siltstone. Its floor is composed of sandy mudstone, medium-fine sandstone, coarse sandstone, siltstone, mudstone, and carbonaceous mudstone. The average thickness of coal seam No.4 is about 2.48 m. Its roof is composed of carbonaceous mudstone, mudstone, sandy mudstone, medium sandstone, and siltstone. Its floor is composed of mudstone, sandy mudstone, siltstone, and calcareous mudstone. The average thickness of coal seam No.5 is about 1.51 m. Its roof is composed of mudstone, sandy mudstone, and carbonaceous mudstone. Its floor is composed of mudstone, sandy mudstone and siltstone. The average thickness of coal seam No.6 is about 1.22 m. Its roof is composed of mudstone, carbonaceous mudstone, sandy mudstone, and medium-fine sandstone. Its floor is composed of mudstone, sandy mudstone, medium-fine sandstone, and siltstone. The average thickness of coal seam No.8 is about 4.04 m. Its roof is composed of mudstone, carbonaceous mudstone, sandy mudstone, and siltstone. Its floor is composed of mudstone, sandy mudstone, and carbonaceous mudstone. The average thickness of coal seam No.9 is about 2.13 m. Its roof is composed of mudstone, carbonaceous mudstone, and sandy mudstone. Its floor is composed of mudstone, sandy mudstone, and medium-fine  sandstone.

According to the lithologic characteristics, there are three types of aquifers in the coalfield. From top to bottom, there are a Quaternary porous phreatic aquifer, a Carboniferous, Permian sandstone fissure confined aquifer, and a Ordovician karst limestone confined aquifer.

The Quaternary phreatic aquifer is composed of sandy clay; clay; and a mixture of silt, pebble, and bedrock fragments. The loose layer on the ridge is thin, permeable, and weak in water yield; the alluvial fan, plain, and valley in the southern part of the minefield are generally medium in water yield, mainly receiving atmospheric precipitation.

The fractured sandstone-confined aquifers consists of two aquifer groups whose details are as follows: As shown in Figure 3, the first part is the Permian clastic sandstone fractured aquifer group, which is located in the coal seam roof of the Shanxi formation. It contains $K_4$ and $K_6$ aquifers. The thickness of $K_4$ medium-grained sandstone in the Shanxi formation is 0.6–12.41 m, with an average of 3.5 m; the thickness of $K_6$ sandstone is 3.9–11.02 m, with an average of 6.52 m. According to the pumping data of $ZK_1$, $ZK_2$, and $ZK_3$, the unit water inflows are 0.5461l, 0.0010l, and 0.00157l/s·m, respectively, and their permeability coefficients are 0.0281, 0.0012, and 0.00791 m/d, respectively. Therefore, it can be assumed that the water yield is weak to medium. The other group is the limestone karst fissure aquifer group of the Taiyuan formation of the Upper Carboniferous system, which is composed of four layers of limestone, $L_1$, $K_2$, $L_4$, and $L_5$, and is the main aquifer group in the mine field. $L_5$ limestone is 3.80 m thick, $L_4$ limestone is 4.30 m thick, $K_2$ limestone is 3.46 m thick, and fractures are developed in each aquifer. The unit water inflow of the $ZK_1$ hydrological borehole is 0.2200l/s·m; according to the data of $ZK_2$ and $ZK_3$ hydrological boreholes, the unit water inflow is 0.000989l and 0.0013l/s·m, respectively, the permeability coefficient is 0.0017 and 0.0030 m/d, respectively, and the water yield is weak to medium. The water quality type is $SO_4 \bullet HCO_3^-$ (K+Na)$\bullet$Ca. The water yield of the Qingjiao fault zone in the south of the mine area is medium, and that of the other parts is weak. According to the water level observation data of borehole 343, the mixed water level elevation of the Taiyuan formation is +811.40 m.

Finally, the karst-confined aquifer is located in the Ordovician limestone with a thickness of more than 120 m. The elevation of the groundwater table varies from +773 to +820 m. Based on the pumping test data of the Ordovician limestone aquifer, the unit water

inflow of the $ZK_1$, $ZK_2$, and $ZK_3$ boreholes is 2.2321, 0.0022 and 0.00102 L/s·m, respectively. Most of the mine field is located in the area rich in weak water, and the Qingjiao fault zone on the southern edge and the adjacent goaf are located in the area rich in strong water. Therefore, when mining the lower coal seam, the groundwater in the karst aquifer may flow to the goaf when there are cracks, and the gushing groundwater will seriously damage the underground mining.

## 4. The Accident Sequence and the Rescue Operations

At 14:00 hours on 22 May 2017, a mine worker, who was working at the site, found that there was water in the work head and continued to dig after draining the water.

At 20:22 hours, the mine worker reported to the dispatching room by telephone; the water gushing from the floor near the working head of the setup entry in the 03304 panel increased, which exceeded the drainage capacity of the working face air pump (12 $m^3$/h).

At 21:45 hours, the deputy chief, who was the deputy chief of the geodesy department, went to the working face to observe the water situation and reported to the mine dispatching center underground that the water output was about 20 $m^3$/h, with a rotten egg smell and suspected goaf water. Subsequently, the middle shift workers went down to the working face as usual.

At 22:00 hours, eleven workers of tunneling team No.1 carried two high-power pumps to the 03304 panel and began to organize drainage.

At 23:38 hours, the nightshift workers were installing a drainage facility when a sudden surge of water rushed into the roadway of the 03304 panel and overthrew the field workers. According to the statistics, 67 people entered the mine on duty, and 56 went up safely. The accident caused 11 people to be trapped inside the mine.

At 01:50 hours on 23 May 2017, four of the trapped men were contacted by telephone from the refuge room in the lane of the 03304 panel (600 m from the water burst point), and the remaining miners were unable to be reached.

After the water inrush accident occurred in Dongyu coal mine, it was reported to Shanxi Meijin Energy Group Co. Ltd., Qingxu County People's Government and other relevant departments.

The People's Government of Shanxi Province promptly set up the "5.22" accident rescue headquarters to mobilize relief materials and dispatch three rescue teams from mines of Taiyuan Municipal Government, Xishan Coal & Power Group, and Fenxi Coal Mine Bureau to carry out rescue.

As of 00:36 hours on 24 May, five survivors were rescued and the other six workers died; at 06:27 hours on 24 May, all six workers that drowned were retrieved from the mine; the rescue work ended.

## 5. Characteristics and Mechanism of the Groundwater Inrush

### 5.1. The Source

According to the on-site investigation by the accident investigation team, the floor damage occurred at 11.4 m behind the heading face of the setup entry of the 0344 panel working face in coal seam No.3, resulting in groundwater inrush, as shown in Figure 4.

The water inrush mainly came from the accumulated water in the goaf of coal seam No.2 after mining. The reasons are as follows: First, according to the results of hydrochemical tests, the groundwater characteristics of water inrush have obvious goaf water characteristics, as shown in Table 1. Secondly, at 23:38 hours on May 22nd, the water inflow suddenly increased to about 500 $m^3$/h, and two days later, the water inflow rapidly decreased to 5 $m^3$/h, which conforms to the change characteristics of goaf water permeability. This conclusion is supported by the following facts: the field investigation of the investigation team found that a gravel accumulation area within a range of 32.5 m appeared behind the water inrush point (Figure 4), and the lithology of the outburst was mainly fragmentary fine sandstone and siltstone, which was basically similar to the roof of coal seam No.2 (Figure 5). The integrated tunneling machine fell into the collapse pit, a small

amount of water flowed out of the roadway floor, and a small amount of bubbles came out, as shown in Figure 6.

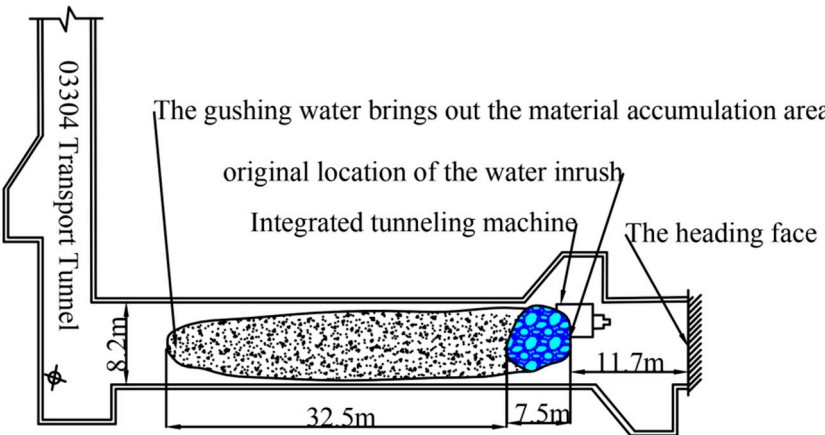

**Figure 4.** Schematic diagram of plane analysis of the water inrush point.

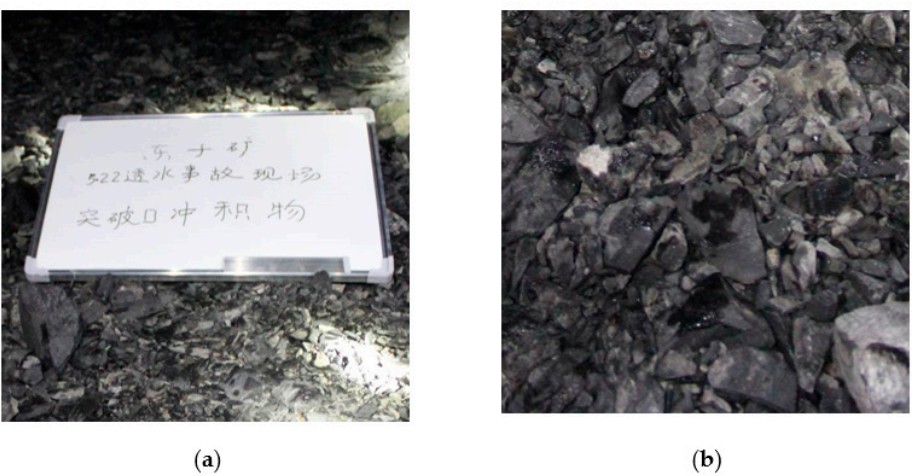

(**a**)        (**b**)

**Figure 5.** Sandy mudstone, siltstone, and other broken rock deposits in the water inrush site. (**a**) Board marked by the investigation team at the scene of the accident; (**b**) gravel with different particle sizes.

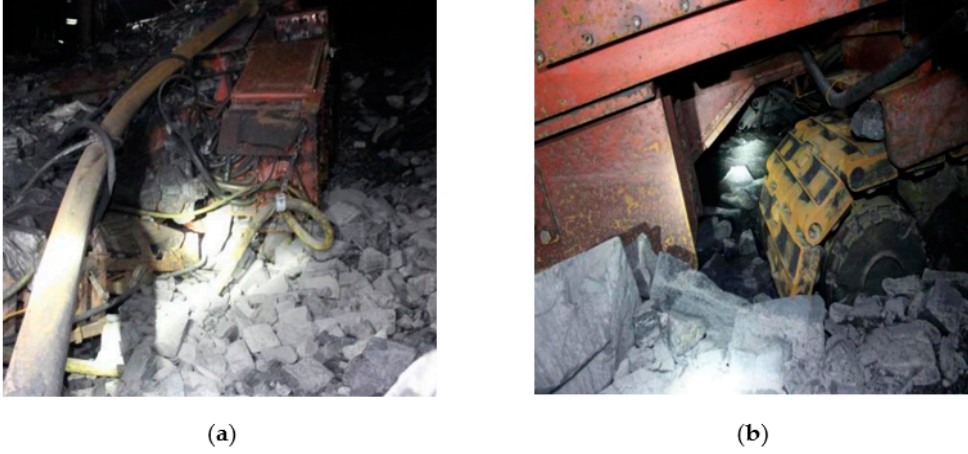

(**a**)        (**b**)

**Figure 6.** A collapse pit formed at the outlet point. (**a**) Power switches and cables that fell into the collapse pits; (**b**) integrated tunneling machine that fell into the collapse crater.

### 5.2. The Passage and the Water Inrush Mechanism

In the middle shift on 22 May 2017, workers found that water began to seep out of the floor. It is inferred that the water-conducting channel began to form at that time. From 20:22 hours to 22:30 hours on the same day, the water inflow increased from 12 to 20 m³/h, which indicates that the water-conducting channel was gradually forming. At 23:38 hours, the water inflow increased sharply with the full formation of water-conducting channel.

As shown in Figures 7 and 8, the fracture zone of the goaf roof of coal seam No.2 in the lower group is the water-conducting channel. The mining disturbance of coal seam No.3 and the high-pressure water in the goaf of coal seam No.2 in the lower group jointly induce the formation of the water-conducting channel.

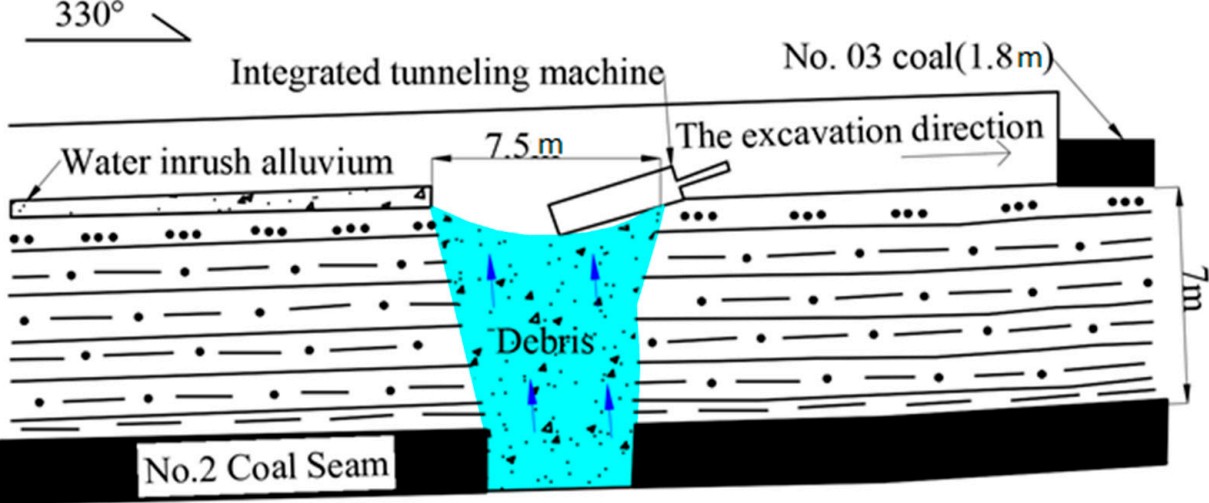

**Figure 7.** Profile diagram of the water inrush location in the heading direction.

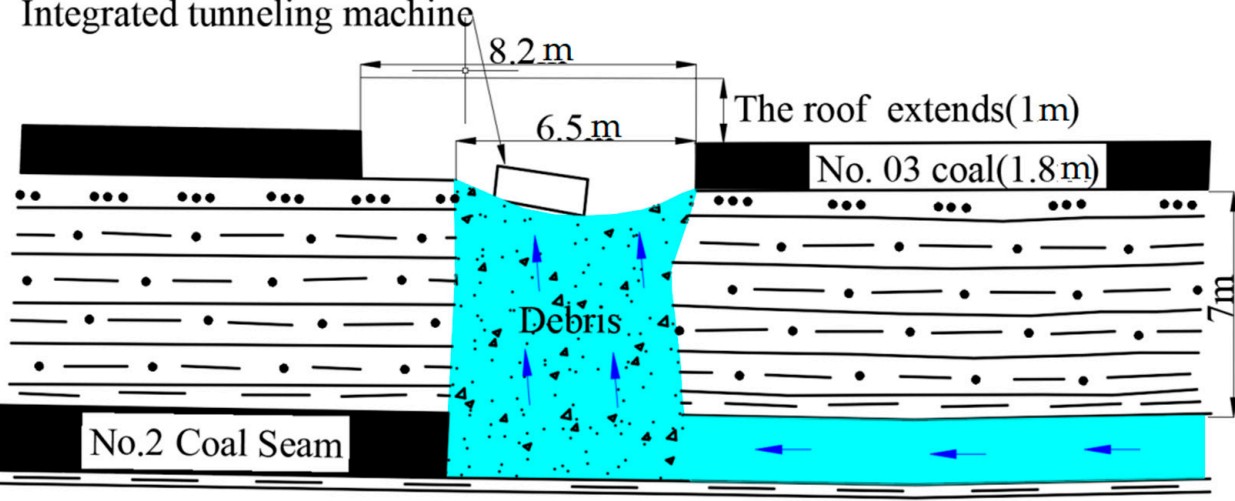

**Figure 8.** Section diagram of the water inrush position perpendicular to the excavation direction.

According to the geological data and borehole data, five boreholes were drilled at the cut opening and 60 m away from the cut opening, respectively, two of which were boreholes in the direction of the floor. The stratum structure and thickness of coal seams No.3 and No.2 were obtained (Figure 9).

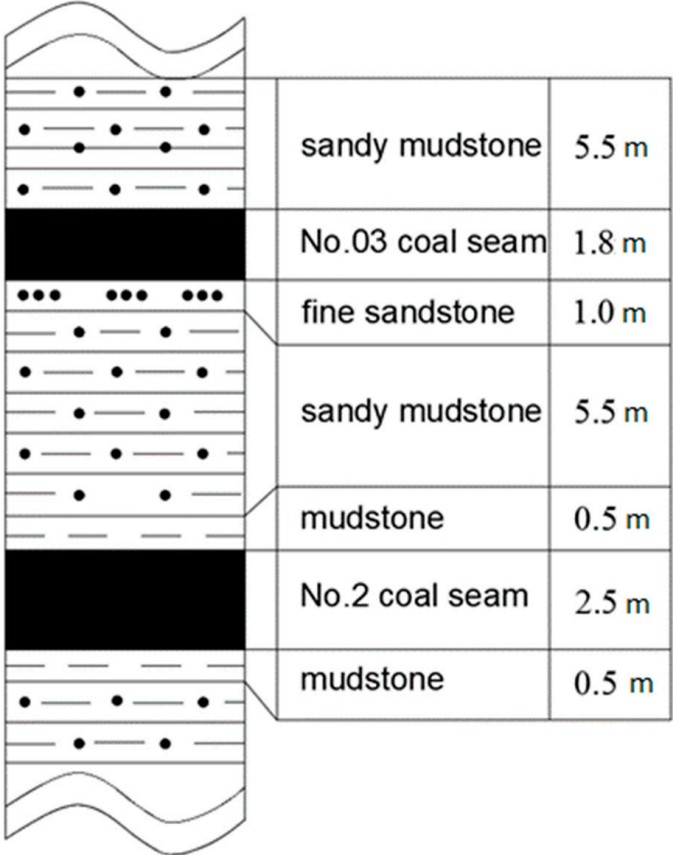

| sandy mudstone | 5.5 m |
| No.03 coal seam | 1.8 m |
| fine sandstone | 1.0 m |
| sandy mudstone | 5.5 m |
| mudstone | 0.5 m |
| No.2 coal seam | 2.5 m |
| mudstone | 0.5 m |

**Figure 9.** Horizon structure of coal seams No.3 and No.2.

The calculation formula of the "Regulations on Coal Mine Water Hazard Controlling" for the thickness of the safe water-resisting layer in the heading face is shown in Equation (1):

$$t = \frac{L\left(\sqrt{\gamma^2 L^2 + 8K_P P} - \gamma L\right)}{4K_P} \tag{1}$$

where $t$ is the safety aquiclude thickness (m), $L$ is width of roadway floor (m), $\gamma$ is the average unit weight of the floor aquifuge (MN/m$^3$), $K_P$ is the average tensile strength of the floor aquifuge (MPa) and $P$ is the actual head value of the water-resisting floor (MPa). The formula of the safe water head of the roadway floor is obtained by Transformation (2):

$$P = 2K_P \frac{t^2}{L^2} + \gamma t \tag{2}$$

The average tensile strength $K_P$ of the waterproof layer of the bottom plate is far less than 2 MPa due to the influence of the nearby collapsed column structure. The effective thickness of the aquifuge between them was reduced to 1–4 m, because the influence of the goaf of coal seam No.2 and the tunneling of coal seam No.3 disturbed the aquifuge of 7 m. The width of the roadway is 8.2 m. The unit weight of the waterproof layer in floor is 0.026 MN/m$^3$. The relationship curve between water pressure and the thickness of the effective insulating layer was obtained as shown in Figure 10.

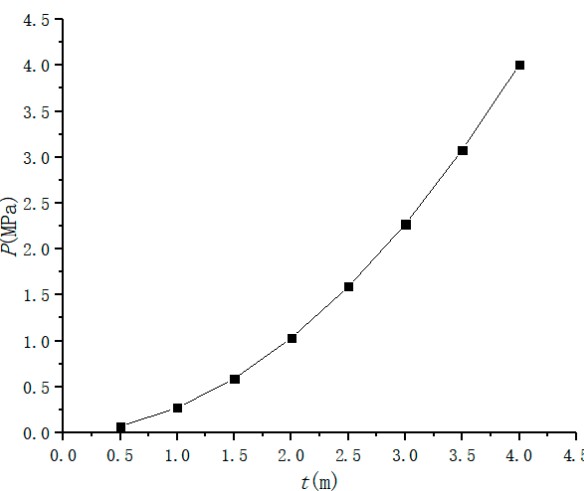

**Figure 10.** Relationship curve between water pressure and thickness of the floor safety barrier.

The actual water pressure is less than or equal to 0.25 MPa, as calculated from the amount of water inrush and the water storage space of the roadway; thus, the thickness of the actual effective waterproof layer is less than 1 m under the influence of mining and other factors. Therefore, the groundwater stored in the goaf of coal seam No.2 poured into the gallery on a large scale, and the disastrous accidents occurred rapidly.

## 6. Causes of the Accident

First, roadway excavation was carried out under the condition that hydrogeological conditions were not fully understood. Relevant geological data, such as the "Mining Engineering Plan & Mine Water Filling Map" used by the miner, fail to truthfully reflect the location distribution of goaf water in coal seam No.2. The figure lacks relevant mining roadway information of coal seams No.3 and No.2 where the accident occurred. In addition, the detailed distribution of goaf water under coal seam No.3 was not found and confirmed in the process of roadway excavation. The accident happened in the setup entry of the 03304 panel, which was located at the synclinal axis, and there was a goaf of coal seam No.2 below it. The water level at the far end of the water-filled goaf was higher than the floor of coal seam No.3, and the floor of coal seam No.3 was under pressure. Under the joint action of water pressure, mining disturbance, and goaf water immersion, the original equilibrium state was broken, and the only 7 m pillar between coal seams No.3 and No.2 was damaged. The water in the goaf of coal seam No.2 rose continuously along the cracks of water-proof rock strata in the roof and gradually changed from seepage to water inrush. In the process, the strength and water-proof ability of the water-proof rock strata were continuously reduced. The water inrush accident happened after the water and gas broke through the rock in the goaf.

Secondly, coal mine management personnel violated the "coal mine water prevention and control regulations" and did not carry out water exploration and drainage in strict accordance with the requirements. Knowing that the working face is close to the boundary of the whole adjacent coal mine, the design of the exploration drilling did not fully consider the detection of the old goaf, and the density of the drilling design was insufficient. Only one floor direction exploration drilling was designed, which failed to effectively explore the underlying coal goaf No.2.

In particular, this factor led to the death of people in the accident disaster. When there was a sign of water gushing at the scene, no instructions were issued to stop the operation and evacuate the personnel immediately. At 14:00 hours on 22 May 2017, water seepage occurred in the floor of the working face. At 20:22 hours, according to the coal mine workers' report, the 03304 panel open-cut roadway was close to the working head position, the floor water of the right-side drilling field became larger, and the water pump

drainage capacity of the working face was insufficient. The workers were asked to stop the excavation operation, but the order to withdraw was not issued. At 21:45 hours, the deputy chief of the geological survey section reported that the water inflow of the mine dispatching center was about 20 m$^3$/h, accompanied by the smell of rotten eggs. It was suspected that it was old empty water, but the those responsible for overseeing operations in the mine still did not withdraw personnel and arranged nightshift personnel to the working face for drainage, which eventually led to casualties in the accident.

## 7. Conclusions and Suggestions

On 22 May 2017, a disastrous groundwater inrush occurred at Dongyu coal mine in Qingxu, Shanxi, China. The accident caused significant losses, including six deaths and direct economic losses of over CNY 5 million.

The groundwater inrush originated from the floor failure in the 03304 appraisal lane in coal seam No.3. The hydrochemical test results of the water inrush source show that the water inrush source was the accumulated water in the goaf of coal seam No.2 under coal seam No.3. It is estimated that the peak discharge of groundwater inflow was 500 m$^3$/h, the total water permeability was 5100 m$^3$, and there was no continuous recharge, which is in line with the characteristics of old empty water.

The results of the follow-up site investigation and related theoretical analysis show that the passage was the fracture produced by the past mining of coal seam No.2 and the stress change in the driving floor of coal seam No.3. The accumulated water in the high-pressure goaf entered the mining roadway through cracks, which promoted the progressive fracture of the floor of coal seam No.3. Finally, a dense channel was formed. The goaf water of coal seam No.2 gushed into the roadway rapidly and strongly.

Tunnel excavation without strict water exploration and drainage measures violates the relevant regulations, resulting in water inrush accidents. In addition, the lack of recognition of groundwater inrush and pervious signs is also one of the reasons for the disastrous consequences of this accident.

Some important lessons can be learned from this event: Some important regulations, for example, "Regulations on Detailed Rules for Coal Mine Water Hazard Controlling" and "Regulations on Water Disaster Prevention and Control of Coal Mine Goaf in Shanxi Province", must be obeyed thoroughly during underground coal mining. It is necessary to strengthen the understanding and prevention of water inrush from the old goaf. Once there are signs of water seepage, corresponding treatments must be carried out on site immediately.

**Author Contributions:** Conceptualization, B.L. and Y.S.; methodology, Y.S., Z.X.; validation, B.L.; formal analysis, G.C., B.L., X.Z. and W.L.; investigation, B.L., H.Y. and X.Z.; writing—original draft preparation, B.L. and W.L.; writing—review and editing, Y.S.; visualization, B.L., L.Z., H.Y.; supervision, Y.S.; project administration, Y.S.; funding acquisition, Y.S. All authors have read and agreed to the published version of the manuscript.

**Funding:** This research was funded by the National Key Research and Development Program of China (No. 2017YFC0804101/No. 2019YFC1805400), the Fundamental Research Funds for the Central Universities (No. 2020ZDPY0201), and the National Natural Science Foundation of China and Shanxi Coal-based Low Carbon Joint Fund Project (U1710253).

**Acknowledgments:** This work was supported by Li Zhenshuan, deputy chief engineer of Taiyuan Coal Geology Bureau of Shanxi Province. The authors would also like to acknowledge the anonymous reviewers for their detailed comments that helped to improve this study.

**Conflicts of Interest:** The authors declare no conflict of interest.

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
