# Peer review of "Damage Characteristics and Mechanism of the 2017 Groundwater Inrush Accident That Occurred at Dongyu Coalmine in Taiyuan, Shanxi, China"

_water, doi:10.3390/w13030368_

Round 1

Reviewer 1 Report

Generally, this technical article is interesting, mostly due to vivid writing style which I appreciate very much. English is fairly good. I would suggest a few revisions in Abstract (a great deal of it should have been written in past tense):

Line 16: The water inrush happened due to following reasons:

Lines 17-22: this is totally confusing me, I can not follow its thread of ideas. Please, revise this (grammar and style). It should be clear and concise.

The other half of the Abstract should be also revised (grammar).

Lines 45-46; data should be given in up to 3 significant digits (2940 died; 67.6%, etc.)

Figure 1: this is in China?

The other parts of the manuscript are quite good, I haven't found any errors or inconsistencies. Minor revision. 

Author Response

Dear Reviewer:

On behalf of my co-authors, we appreciate the opportunity to revise the reviewers' comments on our manuscript “”. We have responded to all comments by the reviewers and made corresponding revisions to the manuscript, as attachment.

Your consideration of this manuscript is highly appreciated. We look forward to your response.

Yours sincerely,

Yajun Sun, Prof.

syj@cumt.edu.cn

Reviewer 2 Report

The manuscript describes a case study of mine water inrush accident. The topic itself is of significance because it caused fatality and major economic losses, and could be interesting for the readers of this journal.

However, the manuscript appears to be poorly written and poorly organised. First, there are numerous incorrect and inconsistent English expressions; secondly the paper is not well organised to provide readers with adequate scientific values. There are superficial judgement of the mechanisms of the water inrush accident without detailed analysis or support from detailed data. It feels more like a management incident investigation report than a scientific paper.

Overall, I suggest the paper needs to be significantly improved or re-written before it can be accepted for publication.

some specific comments:

1. "No.03 Seam. No.2 seam etc.": need to be consistent. No.3 seam?

2. "Comprehensive tunnelling machine": Integrated tunnelling machine?    

3. Do not use individual person's name (e.g. Sheng Minglu, Luo Wu) in the accident description. Just replace them with "a mine worker" or "Deputy chief" et al. This is not an incident report, and should not disclose individual's identity.

4. Figures 8 and 9. What data were these mechanism based on? There is not discussion about the post-accident drilling or investigation of the geology there. 

Author Response

Dear Reviewer:

On behalf of my co-authors, we appreciate the opportunity to revise the reviewers' comments on our manuscript “water-1083095”. We have responded to all comments by the reviewers and made corresponding revisions to the manuscript, as attachment.

Your consideration of this manuscript is highly appreciated. We look forward to your response.

Yours sincerely,

Yajun Sun, Prof.

syj@cumt.edu.cn

Round 2

Reviewer 2 Report

After authors' modifications, I believe most of my previous concerns have been addressed. I agree it is accepted for publication.